# Sustainable Bio-Based Epoxy Resins with Tunable Thermal and Mechanic Properties and Superior Anti-Corrosion Performance

**DOI:** 10.3390/polym15204180

**Published:** 2023-10-21

**Authors:** Rubén Teijido, Leire Ruiz-Rubio, Senentxu Lanceros-Méndez, Qi Zhang, José Luis Vilas-Vilela

**Affiliations:** 1Macromolecular Chemistry Group (LQM), Physical Chemistry Department, Faculty of Science and Technology, University of the Basque Country (UPV/EHU), 48940 Leioa, Spain; ruben.teijido@bcmaterials.net; 2BCMaterials, Basque Center for Materials, Applications and Nanostructures, UPV/EHU Science Park, 48940 Leioa, Spain; senentxu.lanceros@bcmaterials.net (S.L.-M.); qi.zhang@bcmaterials.net (Q.Z.); 3IKERBASQUE, Basque Foundation for Science, Plaza Euskadi, 5, 48009 Bilbao, Spain

**Keywords:** bio-epoxy, soybean oil, tannic acid, mechanical characterization, thermal characterization, anti-corrosion

## Abstract

Bio-based epoxy thermoset resins have been developed from epoxidized soybean oil (ESO) cured with tannic acid (TA). These two substances of vegetable origin have been gathering attention due to their accessibility, favorable economic conditions, and convenient chemical functionalization. TA’s suitable high phenolic functionalization has been used to crosslink ESO by adjusting the −OH (from TA):epoxy (from ESO) molar ratio from 0.5:1 to 2.5:1. By means of Fourier-transform infrared spectroscopy, resulting in thermosets that evidenced optimal curing properties under moderate conditions (150–160 °C). The thermogravimetric analysis of the cured resins showed thermal stability up to 261 °C, with modulable mechanical and thermal properties determined by differential scanning calorimetry, dynamical mechanical thermal analysis, and tensile testing. Water contact angle measurements (83–87°) and water absorption tests (0.6–4.5 initial weight% intake) were performed to assess the suitability of the resins as waterproof coatings. Electrochemical impedance spectroscopy measurements were performed to characterize the anti-corrosive capability of these coatings on carbon steel substrates. Excellent barrier properties have been demonstrated due to the high electrical isolation and water impermeability of these oil-based coatings, without signs of deterioration over 6 months of immersion in a 3.5 wt.% NaCl solution. These results demonstrate the suitability of the developed materials as anti-corrosion coatings for specific applications.

## 1. Introduction

Since the polymeric materials revolution, the petrochemical industry has been, without any doubt, the major source of the plastics manufacturing industry for raw materials and feedstocks, with 85.1% of all plastics being petroleum-derived in 2019 [1]. In the context of an ever-growing society, material consumption and an incipient key resources scarcity, great concerns surrounding all those ubiquitous and non-renewable plastics have emerged. Further, the environmental impact of plastics, releasing greenhouse gasses and harmful substances during their lifespan, and the accumulation of wastes at the end of their use, has led to the pursuit of new alternatives based on environmentally friendly and renewable feedstocks. Among these plastic materials, thermoset resins with highly crosslinked structures, superior mechanical properties, and high thermal, chemical, and dimensional stabilities [2], can find a wide range of applications as end-use products. The use of epoxy thermosets, with over 70% of the global thermoset market, has been growing considerably in recent years. With global market values of over 12 billion USD in 2020 and expected to grow to over 23 billion USD by 2030 at a 7.5% annual growth rate [3], epoxy thermosets can find applicability in paints and coatings (26%), electronics (25%), composites (22%) or adhesives and sealants (20%) to end-use products such as consumer goods (29%), building, construction materials, and machinery (18%), wind turbines (16%), and means of transport (13%), including components for the aerospace industry (7%) [4]. Around 90% of commercially available epoxy resins are fabricated from the reaction between the petroleum-derived and toxic epichlorohydrin (EPC) and bisphenol A (BPA) to obtain the main precursor, the diglycidyl ether of bisphenol A (DGEBA) [5]. Several studies have postulated BPA’s capability to cause endocrine and metabolic disorders, as well as EPC carcinogenesis [6]. Therefore, considering the relevance of these materials in modern society, many research works have been developed to try to enlighten the way for obtaining new epoxy thermosets, ones that are not DGEBA-derived but come from alternative feedstocks with less environmental and health impacts. Such candidates include, among others, vegetable oils (soybean, linseed, canola, sunflower, karanja, etc.). The numerous unsaturations within their triglyceride chains open possibilities for their chemical modification and use as precursors in multi-functionalized materials. One of these possibilities is the obtention of epoxidized derivatives which, in turn, may be used as substrates for the crosslinking reaction [7,8] with different, naturally occurring, hardening substances, such as diamines (i.e., putrescine [9]), diacids (i.e., itaconic acid [10]), polyphenols (i.e., lignin [11] and tannins [12]), or aromatic compounds (i.e., vanillin [13] and eugenol [14]), from which thermostable polymeric networks may be obtained.

The soybean plant “Glycine max” is grown in all tropic and sub-tropic areas with temperate climates preferably between 18 and 35 °C where crop yielding is the highest. With global production of over 350 million tons of beans (2021), it is one of the most important crops to which the world can access (see Appendix A). It is mainly grown as a source of oil and proteins [15].

The dried soybean plant contains an average of 20% of soybean oil. The global production of soybean oil is expected to reach 61.5 thousand tons in 2022/23 [16]. This vegetable oil has a high triglyceride content and is mainly composed of linoleic (18:2, 55%), oleic (18:1, 18%), linolenic (18:3, 13%), palmitic (16:0, 10%), and stearic (18:0, 4%) acids [17], and is applied in soaps, lubricants, paints and coatings, and all kinds of plastics [18]. Its commercially available epoxidized derivative, ESO, is industrially obtained by the reaction between soybean oil and hydrogen peroxide. Thanks to its multifunctionality, easy availability, and low costs, ESO has attracted remarkable attention as a sustainable alternative to fossil feedstocks for its use as an epoxy monomer to synthesize thermoset resins with tailorable specific properties. This product can be found in matrix resins or as a plasticizing additive in composites [19], pressure-sensitive adhesives [20], self-healing [21], shape memory polymers [22], and vitrimers [23].

In this study, tannic acid (TA) has been selected as the hardening substance. This non-toxic, water-soluble, and biodegradable polyphenol is abundantly found in most vascular plant species and has a biological role of protection against infections, insects, and herbivorous attacks [24]. TA is a high molecular-weight molecule (C_76_H_52_O_46_) with a hyperbranched aromatic structure and potentially 25 phenolic−OH reactive groups that may interact with different epoxy groups in ESO. Thus, curing ESO with TA may compensate for the high flexibility of fatty acid chains and produce highly crosslinked materials with improved toughness and high enough glass transition temperatures (T_g_) and moduli to allow their use in diverse applications [25]. Due to the growing need for obtaining long-term health and environmentally sustainable polymeric materials, together with their affordable price, which is comparable to other commercially available hardeners (approx. USD 180/kg) [26,27,28], the scientific and technological communities have focused their attention on this product during the last decades. Apart from its already available common uses as food and pharmaceutical additives and the tanning of leather [29], TA has been recently studied as an antioxidant and antibacterial component in materials for biomedicine [30,31], adhesives [32], as a toughening agent in phenol and epoxy resins [33,34,35], nanomaterial synthesis [36], and as a component in flame retardant materials for their potential release of phenoxy radicals that are able to quench oxygen radicals produced during combustion, together with their high char yielding [37], among others.

The crosslinking architecture of epoxy resins is known to have a great influence on the processing and final properties of resulting thermosets. Thus, in the proposed ESO and TA systems, the hyperbranched and highly functionalized structure of TA may compensate for fatty acid linearity, producing a highly crosslinked thermoset. These densely crosslinked materials are expected to exhibit unique thermo-mechanical and hydrophobic properties not displayed by more structurally linear thermosets [38].

Thus, in this work, a fully bio-based sustainable organic thermoset resin based on ESO and TA (ESOTA) has been investigated, and the curing of ESO via thermally induced chemical crosslinking was optimized. The ratio between ESO and TA and the curing conditions have been optimized to achieve the best mechanical and thermal properties. Thus, different proportions of ESO and TA were used, with the epoxy:−OH molar ratio adjusted from 1:0.5 to 1:2.5 and cured at three different temperatures. It was observed that a slight variation in the ratio of initial components and curing temperatures could result in the thermoset resins having tailored mechanical and thermal properties, thus, they are ad hoc and modulable to meet specific application requirements. For example, they can be used as anti-corrosion coatings on different substrates, as impermeabilizing layers, as binders for ceramic components, or even as glues for different surfaces on which specific thermo-mechanical properties are required.

## 2. Materials and Methods

### 2.1. Materials

Epovinstab H-800 D epoxidized soybean oil (oxirane index ≥ 6.4%) was donated by Hebron S.A., La Llagosta, Barcelona, Spain. Tannic acid (C_76_H_52_O_46_, Mw = 1701.2 g·mol^−1^) and absolute ethanol were purchased from Sigma-Aldrich^®^, St. Louis, MI, USA.

### 2.2. Coating Mixture Preparation

TA was dried in a vacuum at 100 °C for 24 h and then dissolved in absolute ethanol until a homogeneous honey-colored solution was formed. ESO was then added to the TA solutions by adjusting the epoxide: −OH molar ratio, as shown in Table 1. Additionally, samples with ratios of 1:1.66 and 1:5 were prepared to show the effect of introducing additional TA in the TGA−dTG analysis of these materials’ thermal stability. The solution was mixed under magnetic stirring for 2 h and afterward placed under ultrasonic vibration for 30 min. A brown yellowish, highly viscous resin was obtained.

### 2.3. Coating Application and Curing

The resins were cured, either as bulk materials for structural, thermomechanical, and water absorption characterizations, or as coatings on carbon steel plates (DC−01, C% ≤ 0.12, 76 × 25 × 0.8 mm) for water contact angle and electrochemical characterizations. For bulk samples, 5 mL of each mixture were poured on silicon molds, which were then degasified under reduced pressure (−76 mmHg, 4 h) and dried to eliminate EtOH. Afterward, samples were cured in a muffle oven by heating from T = 20 °C to T_curing_ in 18 h (T_curing_ = 100, 130, 170 °C) and maintained at T_curing_ for 4 additional hours. EtOH can be eliminated either during the degasifying process or at the slow heating ramp, which avoids the formation of bubbles and holes in the final cured resins. For coated samples, 1.5 mL of each formulation was deposited on carbon steel substrates with a doctor blade to obtain coating thicknesses of 100 ± 20 µm. Grease-proof paper masks were employed to obtain samples with homogeneous-shaped coated surfaces. The experimental procedure is schematically represented in Figure 1.

### 2.4. Fourier-Transform Infrared Spectroscopy (FTIR)

FTIR transmittance spectra of the samples were recorded between 400 and 4000 cm^−1^ with a resolution of 4 cm^−1^ with a Nicolet Nexus 670 FT−IR spectrometer. All samples were prepared in KBr pellets. Additionally, a series of spectra were recorded when heating an uncured mixture sample from 20 to 170 °C (1 °C·min^−1^) and maintained at 170 °C for four additional hours for a total of 400 min. One spectrum per minute was recorded over the experiment’s total duration.

### 2.5. Thermogravimetric Analysis (TGA) and Differential Scanning Calorimetry (DSC)

The thermal stability of the samples was determined by TGA between 25 and 800 °C with a heating rate of 10 °C·min^−1^ under N_2_ inert atmosphere in a Shimadzu DTG-60 apparatus (Kyoto, Japan). The thermal behavior was also studied by DSC, carried out in a Mettler-Toledo DSC822^e^ calorimeter between 0 and 200 °C (20 °C·min^−1^) under N_2_ atmosphere (Columbus, OH, USA). Thermograms were recorded during sequences of two heating segments intercalated with a cooling segment.

### 2.6. Dynamical Mechanical Thermal Analysis (DMTA)

To characterize the thermo-mechanical properties of the bio-resins, each bulk-cured sample with a thickness of 2.15 ± 0.5 mm was cut down to pieces 5 mm in length and 4–6 mm in width, which was required by the instrument. The DMTA analysis was carried out in a Mettler-Toledo DMA1 Stare System in the tensile setup between −100 and 200 °C with a maximum deformation of 5 µm applied at a 10 Hz frequency.

### 2.7. Tensile Tests

Tensile tests were performed at room temperature with a Metrotec MTE-1 testing machine with a 500 N load cell (TSTM 500 N, class: 00, out: 2 mV/V, AEP Transducers). The initial length of samples was fixed to 9 mm, and the deformation applied was 3 mm·min^−1^ without preload.

### 2.8. Water Contact Angle (WCA)

Water contact angle measurements for determining the wettability of the coated surfaces were performed with a Dataphysics OCA15 EC (Stuttgart, Germany), which comprises an optical contact angle measuring and contour analysis system (sessile drop, 4–6 µL). Results are provided as the average of 7 measurements for each coating.

### 2.9. Water Absorption Tests

Water absorption tests were performed by periodically measuring the weight increase in a Mettler Toledo, NewClassic MF, MS304S model balance due to the absorption of water. For each formula, three cured samples were immersed in distilled water over 90 days.

### 2.10. Electrochemical Impedance Spectroscopy (EIS)

Coatings were immersed in an aqueous NaCl (3.5 wt.%) solution for 24 weeks. During this period of time, their anti-corrosion efficiency was evaluated by measuring their open-circuit potential (OCP) and potentiostatic electrochemical impedance spectroscopy (PEIS) measurements. An IS-CETC PalmSense coating evaluation tests cell and a PalmSense4 portable potentiostat were employed in a 3-electrode setup mode with the coated sample as the working electrode (WE) (Houten, The Netherlands), a platinum wire as the counter electrode (CE), and a Ag/AgCl electrode as reference (RE) with an exposed coated surface of 7.069 cm^2^. The impedance measurements were performed against OCP values, applying an AC potential within a frequency range of 300 kHz to 10 mHz with a sinusoidal perturbation of 10 mV_rsm_ and recording 10 points/decade.

## 3. Results

### 3.1. Structural Characterization and Thermal Curing Optimization by FTIR

FTIR analysis was performed to assess the chemical characteristics of the cured bio-resins. Transmittance spectra recorded for both initial products, ESO and TA, as well as for a sample resin of the stoichiometric 1:1 epoxi:−OH molar ratio (Figure 2 and Figure 3), allowed us to identify characteristic vibration bands in the cured material, as presented in Table 2. A broad band centered around 3432.73 cm^−1^ corresponds to the stretching vibration of unreacted −OH groups present in TA, as well as to the newly formed −OH groups as a result of the epoxide opening reaction during thermal curing. The slight displacement of the −OH band from its position in the TA spectrum (from 3374.87 cm^−1^ to 3432.73 cm^−1^) could be attributable both to the formation of new −OH groups and the formation of hydrogen bonds due to the interaction with surrounding compatible functionalities, such as additional hydroxyl groups or ester carbonyls, present in both components. Two sharp bands appear at 2927.46 cm^−1^ and 2854.18 cm^−1^. They are assigned to symmetric and antisymmetric stretching vibrations for methylene (−C_sp3_−H_2_−) and methyl (−C_sp3_−H_3_) units, which can be attributed to the aliphatic chains in ESO structure at 2925.53 cm^−1^ and 2856.11 cm^−1^, as observed in the corresponding spectra. Additionally, a broadened band at 1733.72 cm^−1^ in the cured resin spectra is found, corresponding to the combination of both ester carbonyl group stretching vibrations, present in the spectra of both initial materials (at 1743.36 cm^−1^ in ESO and 1716.36 cm^−1^ in TA), which remain unreacted during the curing process. Characteristic bands in the tannic acid spectrum centered around 1612.23, 1533.16, and 1446.37 cm^−1^, identified as C_sp2_=C_sp2_ aromatic stretching vibrations, are also observed in the cured resin spectrum at 1614.15, 1535.09, and 1456.02 cm^−1^, showing this last band a broad shoulder due to the combination with the −C_sp3_−H_2_− bending in the ESO spectrum at 1463.73 cm^−1^. In the region between 1322 cm^−1^ and 1081 cm^−1^, several strong bands are observed in all the spectra (Figure 2). These bands are attributable to the O=C_sp3_−O stretching vibration typically found in esters.

Finally, the band at 757.90 cm^−1^ is characteristic of tannic acid C_arom._−H out of plane bending vibration. This band is also observed at 759.83 cm^−1^ in the cured resin spectra, presenting a shoulder at 727.04 cm^−1^ due to the contribution of −C_sp_^3^−H_2_− vibrations at 725.12 cm^−1^.

The bands corresponding to the stretching vibrations of the oxirane −C_sp3_−O−C_sp3_− (antisymmetric and symmetric) were identified at 844.68 cm^−1^ and 823.47 cm^−1^. Figure 3 shows the FTIR spectra of pre- and post-cured resins, in which the disappearance of those bands, as a result of the curing reaction process, is evidenced.

As summarized in Table 3, a shift of the band from 844.68 cm^−1^ to 842.75 cm^−1^ is observed as a consequence of temperature increase, and this peak remains at 842.75 cm^−1^ until its full disappearance between 150 and 160 °C. Meanwhile, as can be observed in the bidimensional mapping (black rectangle) (Figure 4a), the intensity of the transmittance signal for the studied bands becomes higher, initially in a subtler way from green to yellow for lower temperatures. With a more abrupt change to orange at temperatures above 160 °C, the disappearance of the oxirane functional group is evidenced as the curing reaction progresses. These intensity changes appear clearly in the magnified spectral region of interest in Figure 4b. Finally, a displacement of the band at 823.47 cm^−1^ to 829.25 cm^−1^ is observed at temperatures above 145 °C, and the peak does not completely disappear until the end of the thermal curing process. The disappearance of the band at 845 cm^−1^ and the shifting of the band at 823 cm^−1^ are assigned to the end of the reaction, and the remaining band corresponds to the unreacted oxirane groups due to the high steric hindrance imposed by a previous reaction of a neighboring oxirane with a bulky molecule of TA. Thus, the optimal temperature for the crosslinking of both initial products was established between 150 and 160 °C.

### 3.2. Thermal Characterization of the Cured Bio-Epoxy Resins

Thermogravimetric analysis was employed to evaluate the thermal stability of the different cured resins and to study the effect of excess TA on thermal stability. Thermal events registered during the thermogravimetric measurements, obtained from the thermograms in Figure 5a,b, are presented in Table 4. In the TA thermogram (Figure 5b, blue), three thermal events are observed. The first event occurs between 30 and 104 °C and is centered around 66 °C, corresponding to the evaporation of adsorbed ambient water and losing around 10% of its initial mass. The second event takes place from 227 to 350 °C. It is centered around 259 °C and identifies the first decomposition stage of TA in which ester bonds are thermolyzed, losing around an additional 48% of the remaining weight. The thermal decomposition of TA has a second stage from 350 to 600 °C, corresponding to the pyrolytic carbonization of the remaining product by losing approximately 26% of its remaining weight with a carbonaceous residue of 16% of the initial product mass. On the other hand, ESO (Figure 5b, green) is thermally degraded in a single stage, from 325 to 481 °C, losing approximately 84% of its initial mass. A small change in the slope and the identification of two features in the dTG signal between these temperatures indicate two different thermal events during this decomposition, which are attributable to the thermal cleavage of different chemical bonds (ester, oxirane, single aliphatic bonds) within ESO molecules until the carbonization of the sample, leaving 12% of the carbonaceous residue. Thermograms for the cured resins with different ESO:TA ratios (Figure 5a) show one single decomposition step, similar to the one found for ESO, between 231–261 °C and 470–485 °C and centered around 365–378 °C. An increasing shift to higher temperature is found for those samples with less TA proportions, identifying variations in the thermal stability. Thus, a compromise must be reached for each specific application between the required thermal stability and the mechanical properties of the final product, which, as will be discussed later, are significantly affected by the amount of TA employed on each formulation. The dTG curves for the cured resins were found to be composed of at least four contributions, two from TA and two from ESO, with their shapes evidencing the combination of the four contributions. Out of the four contributions, two peaks can be clearly observed. The first one between 365 and 376 °C corresponds to the highest degradation rate for the TA segments within the cured resins, with a tendency toward lower temperatures and higher intensities with the increase in TA content. The second peak appears at between 417 and 421 °C, corresponding to the temperatures of maximum degradation rate for the ESO segments within the matrix, with more stable values due to all formulations containing the same amount of this initial product. The carbonaceous residues values found once the thermal program was finished did not follow the expected trend for higher crosslinked densities leaving higher amounts of carbonaceous residues. For these materials, the carbonaceous residue seems to go to lower values until a 1:1.66 molar proportion of epoxy:−OH is reached. It goes to higher values in coatings with formulations including higher excesses of tannic acid. This may be explained due to TA being a good candidate for obtaining carbonaceous materials after calcination [39,40].

Thus, in formulations with TA equal or slightly higher than the stoichiometric epoxy: −OH proportions and with TA molecules more dispersed through the whole matrix structure, less carbonaceous residue is left. Meanwhile, when a higher TA excess exists, unreacted TA molecules tend to agglomerate during the heating program, thus leaving higher carbonaceous residue, which would be expected from this material alone.

Further, DSC measurements were performed on the different cured resins. Different endothermic transitions were found in the two consecutive heating scans (in Appendix A and Appendix A). In the first scan between 50 and 108 °C, a composed curve containing several components was observed attributable to the evaporation of the residual water in the samples, both superficially adsorbed and incorporated within the resin. This transition is more obvious in the samples with higher TA contents due to the higher water affinity, as shown in the TGA results, showing a 10% weight loss due to water evaporation in pure tannins between 30 and 104 °C. The second scan showed an endothermic transition located between 115 °C for pure tannic acid and 135 °C for the sub-stoichiometric (1:0.5) epoxy:−OH molar ratio formulation of the cured resins. This transition is characteristic of the tannins T_g_ appearing in the same region as different tannin species [41,42]. The descending trend of tannins T_g_, as TA content increases, may be explained due to the plasticizer effect imposed by a higher presence of much bulkier TA molecules in the final thermoset structure. Due to the high reticulation of these amorphous resins, no other significant transitions were observed in the DSC thermograms. Thus, the final T_g_ values for the cured resins were determined by means of DMTA.

### 3.3. Dynamical Mechano-Thermal Characterization

The dynamic mechanical thermal properties were also studied, and Figure 6 shows the thermograms for the storage modulus and tan δ values. The T_g_ values were evaluated for all samples. Every sample from each formulation was cured at three different temperatures (100, 130, and 170 °C) below and above the optimal curing temperature (150–160 °C) determined by FTIR. Values for the determined T_g_ and storage modulus at 25 °C are gathered in Table 5. As expected, an increase in TA content produced an increase in the storage modulus values calculated at 25 °C (see Appendix A for magnification between 0 and 50 °C of the storage modulus thermograms presented in Figure 6), up to the 1:2 epoxy:−OH ratio. Although, in some cases with a 1:2.5 epoxy:−OH ratio, the storage modulus registered was slightly lower, a higher stability of their values was observed, as the moduli decreased more slowly during the heating program between 0 and 50 °C. Meanwhile, higher storage modulus values were obtained on every case for samples cured at higher temperatures, indicating that TA in excess produces the rigidification of these bio-resins.

By analyzing the tan δ curves, two main issues can be assessed. First, an increasing trend of T_g_ values, calculated as the maximum peak for these curves, is observed in the samples with higher TA contents, corresponding to highly crosslinked materials that are in need of higher temperatures to modify their mechanical behavior. Second, by the shape of the curves, a trend to obtain sharper peaks is observed, both at higher curing temperatures and with increasing TA contents, indicating the materials with higher homogeneities on their chemical structures due to a higher degree of crosslinking, on which more similar combinations of applied temperatures and stresses are needed to modify the mechanical response of their whole structure. DMTA analysis allows us to assess the tunability of mechanical properties regarding each specific application, from soft plastic materials with sub-stoichiometric TA proportions (1:0.5) and lower curing temperatures to highly rigid materials as the TA content or curing temperature was increased. A temperature of 170 °C and a 1:1.75 epoxy:−OH molar ratio are the optimal conditions for the best mechanical properties, as no significant improvement was observed in formulations with higher TA ratios. Although stable materials were obtained from the samples (1:0.5) cured at 100 °C, much lower storage modulus and glass transition temperatures were observed in these cases due to their loose crosslinking produced by the deficiency of the hardener substance or the unfinished thermal curing process. The crosslinking density (V_e_) for each formulation of cured resins has been calculated and included in Table 5, employing the rubber state equation (Equation (1)) in the same manner as in previously reported works [43,44].
V_e_ = E′/(3·R·T)(1)
where E′ is the storage modulus corresponding to T, R is the gas constant, and T is the temperature of the rubber state for which T_g_ + 50 °C was considered for each specific formulation. Calculated crosslinking density values followed an increasing trend when higher curing temperatures were employed, except for samples cured at 100 °C, which were poorly crosslinked due to the lack of higher temperatures, as evidenced by much lower V_e_ values one order of magnitude lower than those for samples cured at higher temperatures. An analog increasing trend was also found for higher TA proportions until the 1:1.5 epoxy:−OH molar ratio was reached. Formulations with higher TA proportions show lower crosslinking density values, hinting at the hindering effect on the matrix crosslinking exerted by unreacted bulky TA molecules. An epoxy:−OH 1:1.5 molar ratio was found to be the optimal proportion to achieve matrix structures with higher crosslinking densities for samples cured at 130 and 170 °C, with considerably higher values found in the late ones.

### 3.4. Tensile Tests

Previous works have stated the dependency in the final mechanical properties of a cured epoxy resin on the chemical structure and initial ratio of both hardener and epoxydic components [8,10,12,14] due to the formation of thermoset polymeric networks with different crosslinking densities. For the synthetized resins, this was already evidenced through the DMTA analysis with, as previously discussed, higher temperatures for the tan δ peaks and sharper curves at higher TA contents, indicating higher crosslinking degrees. To further study the mechanical properties of these bio-resins and their variation with different ESO and TA initial ratios, as well as complementary information to the DMTA results, tensile tests were performed on all samples cured at 170 °C. From the obtained stress–strain curves (Figure 7), Young’s modulus, maximum stress, and elongation at break were calculated and are shown in Table 6. According to the obtained values, the maximum tensile strength for the different formulations increases with higher TA content, from the epoxy:−OH ratio 1:0.5 (1.23 MPa) to 1:2.5 (38.6 MPa), meaning an increase in crosslinking densities, which are able to withstand higher stresses without breaking apart, which corroborates the results obtained by DMTA. Meanwhile, the deformation at break first increases, from 1:0.5 to 1:1 formulation, and remains at similar values with increasing TA proportion up to 1:1.5, decreasing for higher TA contents due to the significant change on the reticulation of these systems, which is also evidenced by greater changes on their Tg seen on the DMTA results. As in DMTA, the obtained values for the studied parameters from the tensile tests also exhibit less significant changes in the samples with a ratio of ESO and TA higher than 1:1.5.

The calculated Young’s modulus values for the samples follow a similar trend to maximum stresses and elongations at break, with values from 12 MPa in the sub-stoichiometric sample formulation to between 161 and 232 MPa for the rest of the formulations, again increasing as the TA proportion becomes higher. Although perfectly stable cured resins were obtained from the sub-stoichiometric proportion (1:0.5), this formulation displayed a considerable difference between their mechanical properties and those from formulations over the molar stoichiometry ratios (1:1). Regarding the results obtained from DMTA analysis, although higher crosslinking densities were found for 1:1.5 epoxy:−OH formulation from which a higher Young’s modulus and max. stress values would be expected, it seems that unreacted TA molecules dispersed in the matrix still produce a reinforcing effect against tensile efforts, as evidenced by the higher values obtained during these tests for formulations over 1:1.5. Considering the whole set of results from DMTA and tensile tests, we could state that the adaptability for these bio-resins’ final properties to meet specific requirements for applications were determined by hardness, elasticity, or thermal shock behaviors with a slight variation of both initial components.

Regarding the literature, the determined values for most significative mechanical parameters, Young’s modulus, max. stresses and %deformations, thermal properties, decomposition temperature, and T_g_, were found to be comparable and follow similar trends to those shown by analogous bio-resins made from different epoxidized vegetable oils and tannic acid [7,8]. Looking to go a step forward from these previously reported works, we thought of a direct application for these bio-resins as coatings against corrosion for steel surfaces, with high applicability in marine transportation, wind turbines, and all kinds of offshore infrastructures. Thus, from now on, results regarding the compatibility and behavior of these systems on a water-surrounded media are presented, as well as the initial results for their anti-corrosion capability characterization.

### 3.5. Water Contact Angle

The results obtained from water contact angle measurements for the formulations with different epoxy:−OH ratios are shown in Table 7. From the data presented, a considerable increase in the surface hydrophobicity of the formulated coating with respect to the uncoated steel was observed. The water contact angle was 59.9° for the uncoated steel and ranged from 83.3° to 87.2° for the coated samples. Among the different formulations, an increase in TA proportion due to the hydrophilicity of this component, causes a slight reduction in the surface hydrophobicity of the coated resins with lower water contact angles. Thus, in formulations with higher crosslinking degrees, the affinity for water in the final materials also slightly increased, which could be important for applications requiring that paints both be impermeable and have high mechanical requirements; therefore, a balance between these properties needs to be reached.

### 3.6. Water Absorption Tests

Considering possible application fields for the obtained bio-resins in contact with water or even water-surrounded media, water absorption tests were conducted to further study the capacity and extent of these systems to incorporate water over time. Small amounts of water were incorporated, between 0.6 and 5.6% of the initial weight for 180 days (Figure 8), indicating again that these coatings are water resistant. However, a trend to higher weight gains was observed in samples with higher TA proportions due to the higher hydrophilicity of this component that, as observed in the TGA analysis of a commercial sample, may naturally contain up to 10% of its initial weight in water.

### 3.7. Electrochemical Characterization of the Anti-Corrosion Capability

Finally, as the coatings increase the water contact angle of the substrate and are water resistant, their anti-corrosion capability in marine environments was evaluated. The preliminary EIS measurements were performed on a sample of the stoichiometric molar ratio epoxy:−OH (1:1). EIS results are presented as Bode and Nyquist plots in Figure 9, which were continuously recorded over 24 weeks.

Due to the low permeability to water and electrical isolating capabilities, the electrochemical signal recorded suffered from big noise as the frequency of the applied voltage increased. Thus, for most cases, the results are presented in the Bode plots from 300 kHz to 1 Hz frequencies with those from the last two measurements (22 and 24 weeks of immersion) presenting data until 10 mHz to evidence the tendency displayed by all samples during the 24 weeks. In the same manner, Nyquist plots have been limited to 30 kΩ to avoid highly noisy signals. The impedance values obtained from the Bode plots, 10^8^ Ω (1 Hz) and 10^10^ Ω (10 mHz), and the phase angles between applied voltage and recorded current intensities evidenced high enough values. They are over 80° for higher frequencies and remained over 40° for lowest frequencies during the 24 weeks of sample immersion in the electrolyte, indicating a superior capacitive behavior and isolating capability that can effectively impede the oxidation reaction from occurring, and, therefore, coatings showed no sign of deterioration during the stated period of time. Additionally, this maintained capacitive behavior is evidenced on Nyquist plots by showing curves almost parallel to the *Y*-axis, which are imaginary impedances of the major component for these measurements. For a comparative purpose, an additional Nyquist plot is presented with the measurements performed on an uncoated steel sample, evidencing low impedance behaviors (ca. 20 Ω), which are expected from a naked carbon steel substrate.

## 4. Conclusions

In this research work, the possibility of obtaining fully bio-based epoxy resins with higher renewability and eco-friendliness has been assessed. The epoxy resins are thus based on two vegetable-origin products, tannic acid and soybean oil. It was observed that a slight variation in the ratio of initial components and curing temperatures allows tailoring the mechanical and thermal final properties of the thermoset resins; thus, they are ad hoc modulables that can meet specific application requirements.

The tuneability that can be achieved for these systems could meet application requirements, as anti-corrosion coatings on different substrates, such as impermeabilizing layers, binders for ceramic components, or even glues for different natural surfaces, among others. Through the different experiments performed, it could be assessed how an increase in tannic acid proportions and curing temperatures would produce tougher materials able to withstand external damages without losing protection efficiency, which is key for thermoset resin coatings and paints. The optimal curing temperature for these ESOTA systems was stated at 170 °C. However, tannic acid should not be increased over a 1:1.5, the ratio at which the highest crosslinking densities are achieved, unless the mechanical toughness is a main requirement for the selected application. Although samples with ratios higher than 1:1.5 evidenced improved mechanical properties both in DMTA and tensile tests, the crosslinking density and hydrophobic capabilities were hindered due to the unreacted tannic acid in excess. Thus, novel environmentally friendly and sustainable bio-based epoxy resins have been developed with a strong application potential.

In particular, the water impermeability and low water intake indicate their possible applications as water or anti-corrosion protection coatings for metal substrates in marine environments, which has been demonstrated in the present work. For this specific application, the corrosion analysis should be extended to all initial component ratios to obtain proper conclusions. However, promising results were obtained for the 1:1 ESO and TA formulation. Considering water absorptions and hydrophobic properties for higher TA mixtures, improved protection with better mechanical properties without compromising their anti-corrosion protection efficiency is possible.

## Figures and Tables

**Figure 1 polymers-15-04180-f001:**
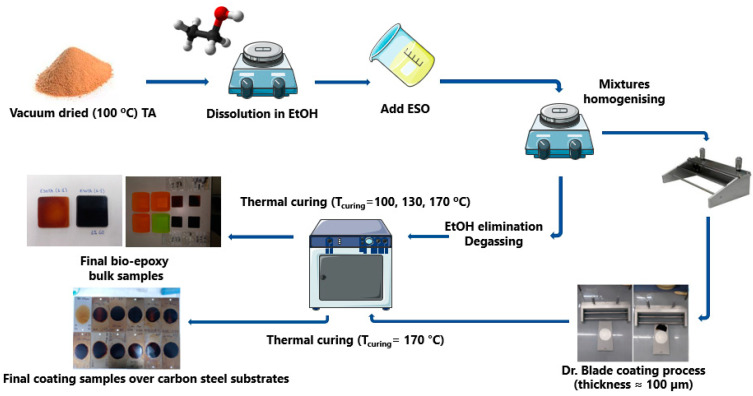
Fabrication process for bulk and coated samples.

**Figure 2 polymers-15-04180-f002:**
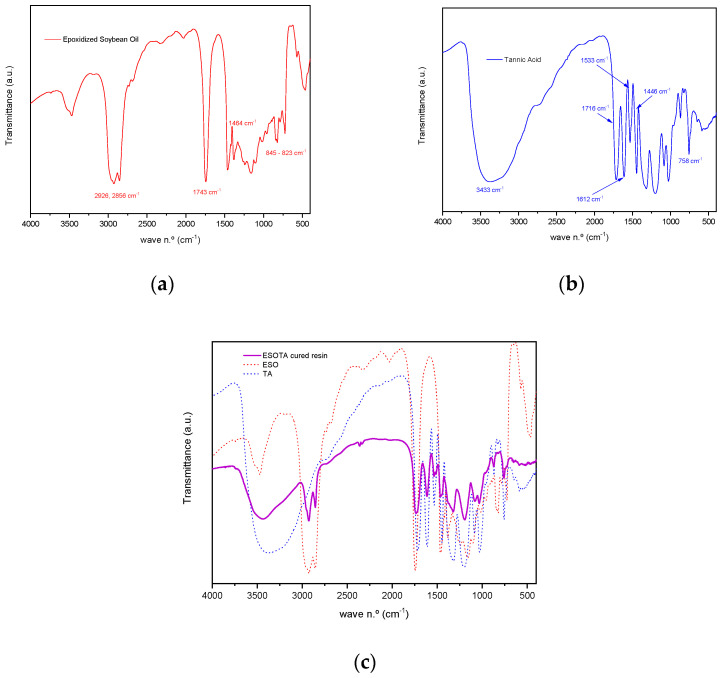
FTIR spectra of the initial components: (**a**) epoxidized soybean oil, (**b**) tannic acid, and (**c**) cured ESOTA resin with a 1:1 epoxy:−OH molar ratio.

**Figure 3 polymers-15-04180-f003:**
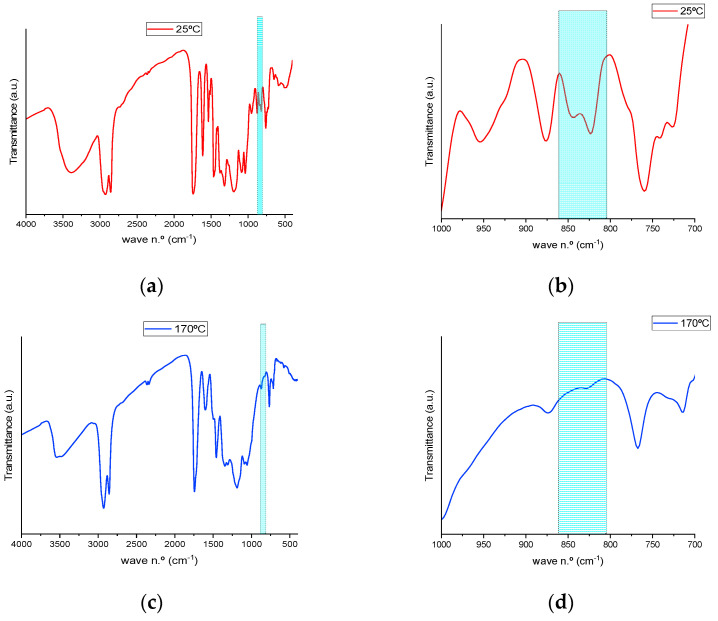
FTIR spectra of the resins recorded before (**a**,**b**) and after (**c**,**d**) thermally induced curing, presented from 4000 to 400 cm^−1^ (**a**,**c**) and from 1000 to 700 cm^−1^ (**b**,**d**).

**Figure 4 polymers-15-04180-f004:**
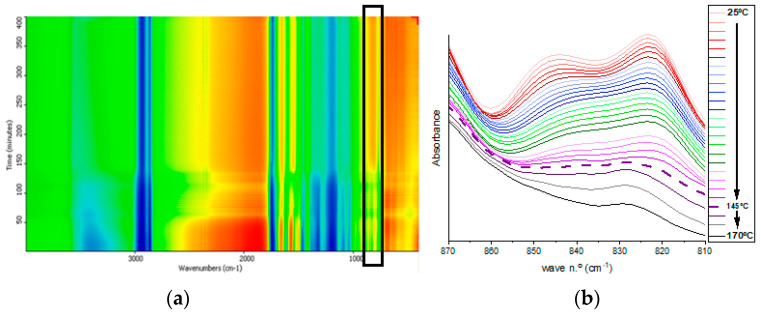
Characterization of the curing reaction. (**a**) Bidimensional mapping of the FTIR spectra recorded during the heating process, epoxide bands are within the black box. (**b**) Evolution of the characteristic epoxide bands in ESO.

**Figure 5 polymers-15-04180-f005:**
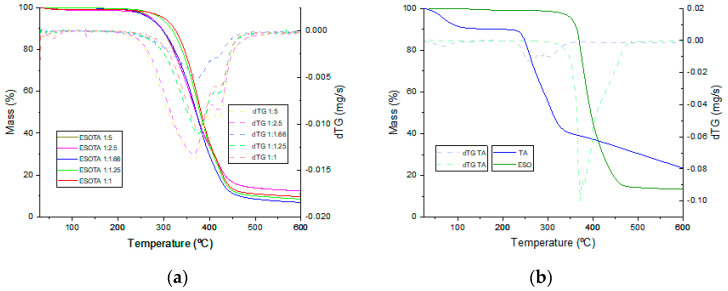
TGA−dTG thermograms of (**a**) formulations with increased TA content and (**b**) of both initial components.

**Figure 6 polymers-15-04180-f006:**
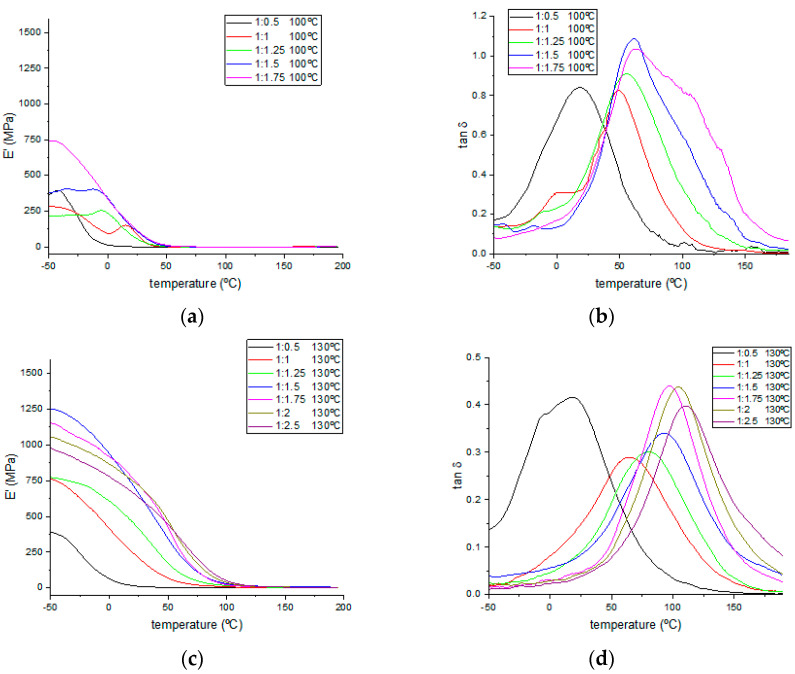
Storage modulus and tan δ curves for each sample cured at (**a**,**b**) 100 °C, (**c**,**d**) 130 °C, and (**e**,**f**) 170 °C.

**Figure 7 polymers-15-04180-f007:**
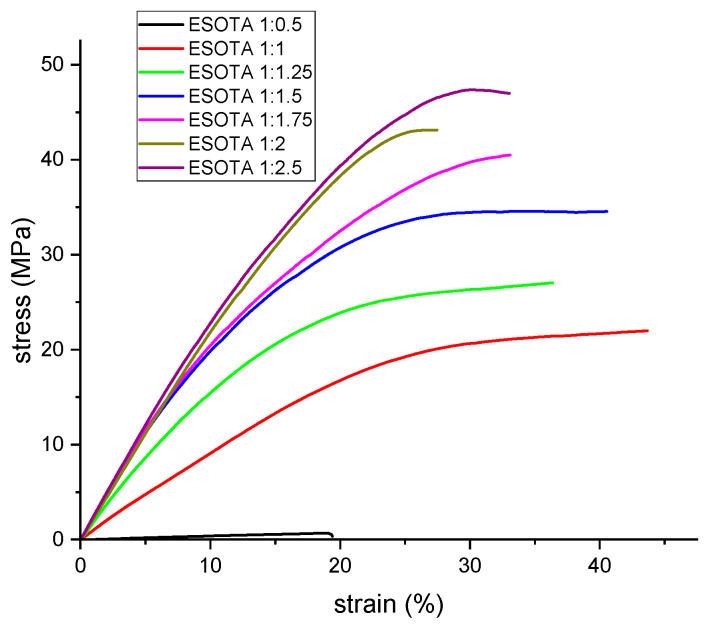
Representative stress–strain curves for the prepared resins with different epoxy:−OH molar ratios.

**Figure 8 polymers-15-04180-f008:**
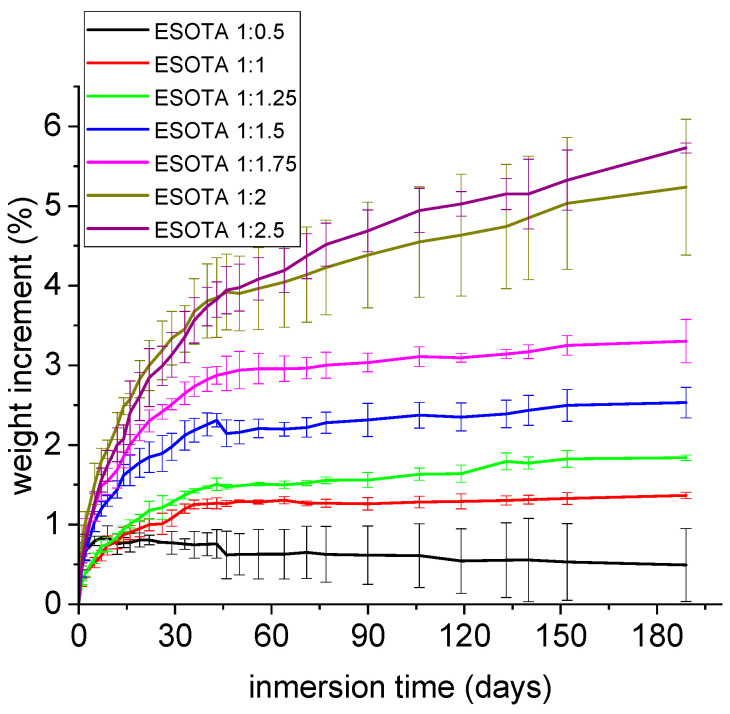
Water absorption results for cured resins with different epoxy:−OH ratios over 190 days of immersion.

**Figure 9 polymers-15-04180-f009:**
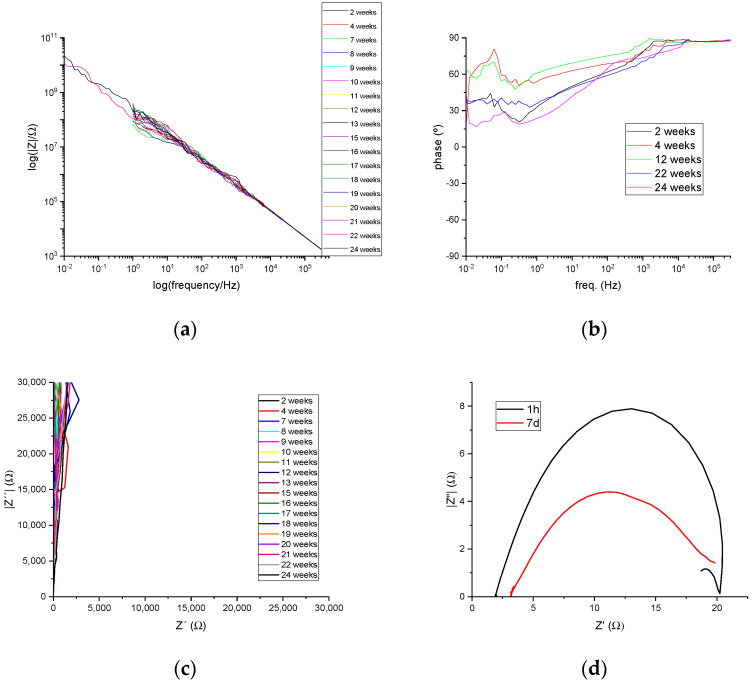
(**a**) Bode plots, (**b**) phase angle plots, and (**c**) Nyquist plots of a 1:1 ESOTA coating sample, and (**d**) Nyquist plots of an uncoated steel substrate sample for comparison.

**Table 1 polymers-15-04180-t001:** Epoxide: −OH molar ratios employed in the different ESOTA prepared bio-resin samples.

Epoxide Groups in ESO (Molar Ratio)	−OH Phenolic Groups in TA (Molar Ratio)	Sample Naming
1	0.5	ESOTA 1:0.5
1	1	ESOTA 1:1
1	1.25	ESOTA 1:1.25
1	1.5	ESOTA 1:1.5
1	1.75	ESOTA 1:1.75
1	2	ESOTA 1:2
1	2.5	ESOTA 1:2.5
1	1.66 (TGA−dTG)	ESOTA 1:1.66
1	5 (TGA−dTG and Contact Angle)	ESOTA 1:5

**Table 2 polymers-15-04180-t002:** Characteristic FTIR vibration bands in ESO, TA, and ESOTA.

Vibration Band	Wave Number (cm^−1^)—ESO	Wave Number (cm^−1^)—TA	Wave Number (cm^−1^)—Cured Resins
ν_symm._ + ν_asymm._ H−O−H	−	3433	3375
ν_symm._ + ν_asymm._ C_sp3_−H	2926, 2856	−	2927, 2854
ν C=O	1743	1716	1734
ν C_sp2_=C_sp2_ (arom.)	−	1612, 1533, 1446	1614, 1535, 1456
δ_symm._ C_sp3_−H	1464	−	1456 (shoulder)
ν_symm._ + ν_asymm_ O=C_sp3_−O−	1241–1105 (triglyceride esters)	1319–1085 (TA esters)	1322–1081 (triglyceride + TA esters)
ν_symm._ + ν_asymm_ −CH_2_−O−CH_2_− (epox.)	845–823		844–823 (uncured mixture)
δ_oop_ C_arom._−H	-	758	760

**Table 3 polymers-15-04180-t003:** Position of the oxirane bands during the curing process in the recorded FTIR spectra.

Sample Temperature (°C)	v_symm._ (C−O−C) Epoxide Bands Wave Number (cm^−1^)
25–30	844.68/823.47
35–130	842.75/823.47
135–155	842.75/829.25
160–170	829.25

**Table 4 polymers-15-04180-t004:** Thermal events registered in the TGA analysis of the initial products and the cured resins with increasing TA content.

Sample	T_i_ (°C)	T_peak_ (°C)	T_f_ (°C)	Weight Loss (%)	Carbonaceous Residue (%)
ESO	25		325	≤2	12
	325	372	481	86	
	481		600	88	
TA	30	66	104	10	16
	227	259	349	58	
	349		600	84	
ESOTA 1:1	25		262	≤2	7
	262	377	485	89	
	485		600	93	
ESOTA 1:1.25	25		264	≤2	6
	264	374	478	89	
	478		600	94	
ESOTA 1:1.66	25		239	≤2	4
	239	375	476	91	
	476		600	96	
ESOTA 1:2.5	25		233	≤2	10
	233	365	474	85	
	474		600	90	
ESOTA 1:5	25		231	≤2	11
	231	365	470	86	
	470		600	89	

**Table 5 polymers-15-04180-t005:** Results for T_g_, storage modulus, and crosslinking densities obtained from the DMTA analysis of samples with different initial component ratios cured at different temperatures.

Epoxide: −OH Molar Ratio	T_curing_ (°C)	T_g_ (°C)	E′ (25 °C, MPa)	E′ (T_g_ + 50 °C, MPa)	V_e_ (mol·m^−3^)
1:0.5	100 °C	18	0.8	0.18	21.18
1:1		49	44.1	0.34	36.67
1:1.25		55	58.4	0.23	24.41
1:1.5		62	105.8	0.18	18.76
1:1.75		63	114.8	0.17	17.67
1:2		*	**	*	*
1:2.5		*	**	**	**
1:0.5	130 °C	17	11.1	2.47	291.45
1:1		64	207.7	3.99	413.63
1:1.25		81	410.4	7.06	701.09
1:1.5		92	664.5	9.27	896.15
1:1.75		97	723.5	5.71	545.43
1:2		105	713.8	5.32	498.67
1:2.5		111	632.8	4.05	374.38
1:0.5	170 °C	17	19.7	5.13	605.32
1:1		76	507.1	6.13	616.36
1:1.25		79	580.9	9.83	981.02
1:1.5		95	806.0	12.72	1220.84
1:1.75		95	826.3	9.89	949.22
1:2		105	732.1	7.33	687.08
1:2.5		106	1159.3	7.10	663.97

*, **: TA in large excess and low curing temperature produced extremely brittle materials that could not be measured.

**Table 6 polymers-15-04180-t006:** Mechanical characterization results by tensile testing for the different epoxy:−OH ratio formulations.

Epoxide:−OH Molar Ratio	T_curing_ (°C)	Young’s Modulus (MPa)	Max. Stress (MPa)	Max. Deformation (%)
1:0.5	170 °C	11.8 ± 2.5	1.2 ± 0.2	9.6 ± 0.9
1:1		161.5 ± 44.6	20.9 ± 3.6	24.8 ± 11.9
1:1.25		184.4 ± 34.2	28.8 ± 2.5	30.1 ± 4.1
1:1.5		194.5 ± 54.9	30.1 ± 3.1	31.0 ± 7.8
1:1.75		208.1 ± 27.4	33.7 ± 7.4	27.8 ± 4.8
1:2		221.6 ± 29.4	36.5 ± 1.9	25.8 ± 10.3
1:2.5		232.0 ± 15.8	38.6 ± 11.3	27.3 ± 22.5

**Table 7 polymers-15-04180-t007:** Water contact angle results and representative images for formulations with different epoxy:−OH molar ratios coated over carbon steel substrates.

Sample	WCA (°)	Deviation (°)	Image
Uncoated steel	59.9	3.8	* 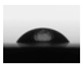 *
ESOTA 1:1	87.2	0.6	* 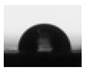 *
ESOTA 1:1.25	85.9	1.1	* 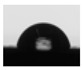 *
ESOTA 1:1.75	84.9	1.8	* 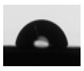 *
ESOTA 1:2.5	84.3	0.6	* 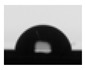 *
ESOTA 1:5	83.3	1.4	* 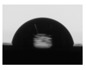 *

## Data Availability

Not applicable.

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
