# Peer review of "Sustainable Bio-Based Epoxy Resins with Tunable Thermal and Mechanic Properties and Superior Anti-Corrosion Performance"

_polymers, 2023, doi:10.3390/polym15204180_

Round 1
Reviewer 1 Report
The article entitled "Sustainable Bio-based Epoxy Resins with Tunable Thermal and Mechanic Properties and superior Anti-Corrosion Performance" deals with use of tannic acid as additives in ESO. Such field is of industrial interest.
The authors describes well the subject of the article, and show the different factors they will study.
The experiment are well described, and the results well presented.
Maybe the figure 2 can be divided in more figure, to avoid a big and heavy figure, difficult to understand.
For me with the small change, the article can be publish.
Author Response
Reviewer 1
The article entitled "Sustainable Bio-based Epoxy Resins with Tunable Thermal and Mechanic Properties and superior Anti-Corrosion Performance" deals with use of tannic acid as additives in ESO. Such field is of industrial interest.
The authors describes well the subject of the article, and show the different factors they will study.
The experiment are well described, and the results well presented.
Maybe the figure 2 can be divided in more figure, to avoid a big and heavy figure, difficult to understand.
As suggested by the Reviewer 1, the graph in figure 2 has been divided in three different FTIR graphs. Two for both initial products (2.a Epoxidized Soybean Oil and 2.b Tannic Acid) and a third graph (2.c) with the spectra for ESOTA 1:1 cured resin highlighted over the spectr for the initial products.
For me with the small change, the article can be publish.
Reviewer 2 Report
The article is well written, easy to follow, with enough examples that the authors test and compare, there are sufficient measurements to enhance the potential of these mixtures. The literature is on topic and updated. I recommend the acceptance of the publication after minor revision.
In the introduction you could insert the following citations based on the importance of epoxides, tannic acid or similar materials as you did:
https://pubs.acs.org/doi/full/10.1021/bk-2021-1385.ch001
Tannic acid as biobased flame retardants: A review – DOI 10.1016/j.jaap.2023.106111
Lines 16-17:
“TA´s suitable high phenolic functionalization has been used to crosslink ESO by adjusting epoxy:-OH molar ratio” – is more correct OH (from TA): and epoxy from ESO, if you start the sentence with the TA’s.
Line 70:
“(see additional information, Figure A1),” – you don’t have Figure A1
Paragraph 118-126 with the Coating mixtures preparation
- You should introduce here the Table with ESO:TA ratios, that you introduce first in Figure 5/graph a, here is more visible and you talk in all article about this mixtures
Line 131:
5mL of each formulation were poured – is sound better 5ml of each mixture not formulation
Paragraph with Results, starting with Line 190
- Is really nice and clear described the FTIR part but you missed to write for which molar ration between ESO:TA you made this study, and is important to know
Table 1. Characteristic FTIR vibration bands in ESO, TA and ESOTA
- You should add an additional row for the cured resin with the wavenumber specific for the ester formation, which you actually want to prove, as you described in Line 213-215 “between 1380 213 cm-1 and 1050 cm-1” “attributable to the O=Csp3-O stretching vibration typically found in esters”
In the final conclusions you presented more general but you have also specific results for each measurement that you could introduce after this paragraph “It was observed that a slight variation on the ratio of initial components and curing temperatures allows to tailor the mechanical and thermal final properties of the thermoset resins, thus, being ad hoc modulable to meet specific application requirements.” For example, which one from all mixtures ESOTA is the best candidate for mechano-thermal properties, for stresses and elongations at break, the trend in hydrophilicity (TA increase slightly hydrophilic) and corelated with water abortion, or the electrochemical characterization of the anti-corrosion capability. Basically, you have all these results after each paragraph with the measurement but you don’t have it in the conclusion.
Author Response
The article is well written, easy to follow, with enough examples that the authors test and compare, there are sufficient measurements to enhance the potential of these mixtures. The literature is on topic and updated. I recommend the acceptance of the publication after minor revision.
In the introduction you could insert the following citations based on the importance of epoxides, tannic acid or similar materials as you did:
https://pubs.acs.org/doi/full/10.1021/bk-2021-1385.ch001
Tannic acid as biobased flame retardants: A review – DOI 10.1016/j.jaap.2023.106111
The suggested references have been introduced in a new paragraph between lines 100 and 108 in which, as suggested by reviewer 3 the motivation of ESO-TA system’s design is stated more clearly due to these 2 products high compatibility and complementary capabilities.
Lines 16-17:
“TA´s suitable high phenolic functionalization has been used to crosslink ESO by adjusting epoxy:-OH molar ratio” – is more correct OH (from TA): and epoxy from ESO, if you start the sentence with the TA’s.
The sentence at lines 16-17 has been corrected to “TA´s suitable high phenolic functionalization has been used to crosslink ESO by adjusting -OH (from TA):epoxy (from ESO) molar ratio from 0.5:1 to 2.5:1.”
Line 70:
“(see additional information, Figure A1),” – you don’t have Figure A1
The reference to Figure A1 was a mistake from an older document version. It has been changed to Figure S1 (line 70-71) which is included in the supplementary information.
Paragraph 118-126 with the Coating mixtures preparation
- You should introduce here the Table with ESO:TA ratios, that you introduce first in Figure 5/graph a, here is more visible and you talk in all article about this mixtures
A table with the employed -OH:epoxide molar ratios used in the different resins mixtures has been introduced after line 137 to clarify the different formulations and abbreviations employed during the whole manuscript.
Line 131:
5mL of each formulation were poured – is sound better 5ml of each mixture not formulation
Suggested change has been made, now standing on line 151 to “5ml of each mixture…”
Paragraph with Results, starting with Line 190
- Is really nice and clear described the FTIR part but you missed to write for which molar ration between ESO:TA you made this study, and is important to know
The ratio between ESO:TA on which the FTIR analysis has been made has been stated in line 214.
Table 1. Characteristic FTIR vibration bands in ESO, TA and ESOTA
- You should add an additional row for the cured resin with the wavenumber specific for the ester formation, which you actually want to prove, as you described in Line 213-215 “between 1380 213 cm-1 and 1050 cm-1” “attributable to the O=Csp3-O stretching vibration typically found in esters”
An additional row has been added to Table 2 (line 247) describing the position for characteristic ester bands found in both initial products and the cured resin.
In the final conclusions you presented more general but you have also specific results for each measurement that you could introduce after this paragraph “It was observed that a slight variation on the ratio of initial components and curing temperatures allows to tailor the mechanical and thermal final properties of the thermoset resins, thus, being ad hoc modulable to meet specific application requirements.” For example, which one from all mixtures ESOTA is the best candidate for mechano-thermal properties, for stresses and elongations at break, the trend in hydrophilicity (TA increase slightly hydrophilic) and corelated with water abortion, or the electrochemical characterization of the anti-corrosion capability. Basically, you have all these results after each paragraph with the measurement but you don’t have it in the conclusion.
As suggested by 2 of the referees, conclusions have been improved, between line 569 and 588 to state some of the trends observed with the results of the different characterization experiments.
Reviewer 3 Report
Currently, on the market of bio-based plastics, the thermoplastics such as PHB, PLA, PCL dominate to implement in construction engineering, packaging, biomedicine and environmental safety areas. Formulation of novel thermoset resins on the base of naturally-originated compounds, in particular ESO cured with TA, is decisively the actual issue, the solution of which is of practical importance for modern materials science.
To reach the realistic findings in thermophysical, mechanical, electrochemical, and water adsorption features for ESO-TA system the traditional methods such as FTIR (regular or 2D imaging), DMTA, tensile technique and the others have been used for the surfaces or bulk characterization with different combination of the components. The content of this paper falls within the scope of the Polymers spanning the biopolymer area. With appropriate terminology and reasonable argumentation, the manuscript shows clearly the effect of hydrophilic curing agent on the complex of above-mentioned physical parameters. The abstract mostly reflects the general issues of the manuscript. The literature cited is quite relevant to this study but the introduction should be expanded to underline the motivation of ESO-TA system’s design. The most of illustrations (the tables and the figures) are executed in unambiguous and accurate manner with the clear coherent interpretation, nevertheless several figures should be improved, please see beneath.
To unify the form of the Title, the adjective “superior” should be started with a capital letter.
The authors are invited to enhance the final section “Conclusions” to elucidate or, at least, discuss the reason of the impact of the small variation ratio (ESO/TA) on thermophysical, mechanical, and barrier behaviors of the fully bio-based films/coatings.
The statistical assessment should be included in the section 2, Materials and methods. The deviations presented in Table 6 and the bars in Fig. 8 should be statistically justified.
It is worth making the rearrangement of Fig.9 to avoid the empty part of the image (c) and simultaneously to enlarge the positions and resolution of corresponding curves.
P 7, L 232 shift of the band at 844.68 cm-1 to 842.75 cm-1 is observed as a consequence of temperature increases, please modify the phrase in such manner.
Only minor amendmentsa are required.
Author Response
Currently, on the market of bio-based plastics, the thermoplastics such as PHB, PLA, PCL dominate to implement in construction engineering, packaging, biomedicine and environmental safety areas. Formulation of novel thermoset resins on the base of naturally-originated compounds, in particular ESO cured with TA, is decisively the actual issue, the solution of which is of practical importance for modern materials science.
To reach the realistic findings in thermophysical, mechanical, electrochemical, and water adsorption features for ESO-TA system the traditional methods such as FTIR (regular or 2D imaging), DMTA, tensile technique and the others have been used for the surfaces or bulk characterization with different combination of the components. The content of this paper falls within the scope of the Polymers spanning the biopolymer area. With appropriate terminology and reasonable argumentation, the manuscript shows clearly the effect of hydrophilic curing agent on the complex of above-mentioned physical parameters. The abstract mostly reflects the general issues of the manuscript. The literature cited is quite relevant to this study but the introduction should be expanded to underline the motivation of ESO-TA system’s design. The most of illustrations (the tables and the figures) are executed in unambiguous and accurate manner with the clear coherent interpretation, nevertheless several figures should be improved, please see beneath.
To unify the form of the Title, the adjective “superior” should be started with a capital letter.
The correction on the title has been introduced changing “superior…” to “Superior…”
The authors are invited to enhance the final section “Conclusions” to elucidate or, at least, discuss the reason of the impact of the small variation ratio (ESO/TA) on thermophysical, mechanical, and barrier behaviors of the fully bio-based films/coatings.
Conclusions have been improved, between line 569 and 588 to state some of the trends observed with the results of the different characterization experiments.
The statistical assessment should be included in the section 2, Materials and methods. The deviations presented in Table 6 and the bars in Fig. 8 should be statistically justified.
Despite the suggestion from the referee, the statistical assessment is not a tool commonly used in neither mechanical characterization and water absorption. Usually, one the article is devoted to biomaterials, including biological responses these analysis are added since they provide an interesting perspective. However, this is not the case. The accuracy of the obtained results is often presented (when possible) as the standard variation calculated from the mean value of different sample replicas, as is the case of the ones presented in table 6 as numerical values and in figure 8 as the error bars on each measurement. Table 6 results were calculated as the mean value for 5 different replicas of each sample formulation while in figure 8 is calculated out of 3 replicas for each sample.
It is worth making the rearrangement of Fig.9 to avoid the empty part of the image (c) and simultaneously to enlarge the positions and resolution of corresponding curves.
When plotting Nyquist diagrams obtained from impedance measurements by EIS it is important to maintain both plot axis with the same scale, as the shape of the curves may give information of the different electrochemical processes happening on the sample. In this case the resulting curves are shown as straight up-forwarding lines evidencing a fully capacitive effect on the polymer-metal interphase, on which the main component for the measured impedance is mostly imaginary (Z’’, Y axis) as stated on lines 539-540.
P 7, L 232 shift of the band at 844.68 cm-1 to 842.75 cm-1 is observed as a consequence of temperature increases, please modify the phrase in such manner.
The sentence now standing in lines 256-257 has been modified in the manner suggested.